# Effects of Ciprofloxacin Alone or in Mixture with Sulfamethoxazole on the Efficiency of Anaerobic Digestion and Its Microbial Community

**DOI:** 10.3390/antibiotics11081111

**Published:** 2022-08-17

**Authors:** Valentina Mazzurco Miritana, Luisa Patrolecco, Anna Barra Caracciolo, Andrea Visca, Flavia Piccinini, Antonella Signorini, Silvia Rosa, Paola Grenni, Gian Luigi Garbini, Francesca Spataro, Jasmin Rauseo, Giulia Massini

**Affiliations:** 1Department of Energy Technologies, Italian National Agency for New Technologies, Energy and Sustainable Economic Development (ENEA), Via Anguillarese 301, 00123 Rome, Italy; 2Water Research Institute—National Research Council (IRSA-CNR), SP 35d, km 0.7, Montelibretti, 00010 Rome, Italy; 3Institute of Polar Sciences—National Research Council (ISP-CNR), SP 35d, km 0.7, Montelibretti, 00010 Rome, Italy

**Keywords:** cattle manure, antibiotic residues, biogas production, antibiotic removal, microbial community composition

## Abstract

Some livestock farms rely on anaerobic digestion (AD) technology for manure disposal, thus obtaining energy (biogas) and fertilizer (digestate). Mixtures of antibiotics used for animal health often occur in organic waste and their possible synergistic/antagonistic effects on microorganisms involved in AD are still poorly studied. This work focuses on the effects of adding ciprofloxacin, alone (5 mg L^−1^) and in combination with sulfamethoxazole (2.5–5–10 mg L^−1^), on AD efficiency and microbial community structure. The experiment consisted of 90-day cattle manure batch tests and antibiotic removal percentages were assessed. Adding antibiotics always promoted CH_4_ and H_2_ production compared to untreated controls; however, CH_4_ production was lowered with the highest ciprofloxacin (CIP) concentrations. The overall results show antibiotic degradation caused by acidogenic *Bacteria*, and CH_4_ was mainly produced through the hydrogenotrophic-pathway by methanogenic *Archaea*. Shifts in microbial community abundance (DAPI counts) and composition (Illumina-MiSeq and FISH analyses) were observed.

## 1. Introduction

An increasing number of intensive livestock farms are relying on anaerobic digestion (AD) technology to solve the disposing of the huge amount of manure and slurry produced daily [1], getting the benefit of energy and fertilizer from biogas and digestate, respectively [2]. The biogas obtained, generally containing 50–75% CH_4_, can be burned in co-generation engines to obtain electric energy [3] or purified to biomethane [4]. The digestate remaining after AD treatment is commonly spread on the soil as a replacement for conventional mineral fertilizers. In the last decade, AD treatment has also been attracting interest for its potential to degrade some organic pollutants and emerging contaminants like pharmaceuticals commonly found in animal manure [5,6], although much research is needed to verify the effectiveness of the process. Veterinary antibiotics are widely used to treat and prevent animal diseases in husbandry practices. Despite their undoubted benefits, their massive use is leading to several risks for environmental processes and ecosystem services [7,8]. Moreover, inhibition of engineered bioprocesses for slurry disposal, such as AD [1], is also suspected. Animals excrete the largest amount (between 10% and 90%) of the antibiotics administered, most of which in unchanged form or as still metabolically active metabolites [9,10,11]. Livestock manure is commonly applied to land as an organic fertilizer, and it is generally recognized that by adding it, antibiotic residues from treating cattle infections can be introduced into soils [12,13]. Sulfonamides, fluoroquinolones, tetracyclines and macrolides are the antibiotics most commonly used in veterinary practices [5,11,14]. Among them, ciprofloxacin (CIP) and sulfamethoxazole (SMX), belonging to the class of fluoroquinolones and sulfonamides, respectively, are the most frequently detected in manure and consequently in agroecosystems [15,16,17]. CIP and SMX differ in their environmental mobility and persistence [5]. CIP shows a high persistence in soil and water [18] due to its recalcitrance to biodegradation [19] and high affinity for the soil matrix. CIP is detected in the environment both because it is widely used also in human medicine and because it is the primary degradation product of enrofloxacin (ENR), a veterinary fluoroquinolone. Moreover, it was found that much of ENR is metabolically converted to CIP [20,21]. SMX shows a low soil adsorption coefficient due to its polar nature [22] and can be biodegraded under aerobic/anaerobic conditions [12,23].

CIP and SMX also show different patterns of antimicrobial activity. The first exerts a bactericidal action through the inhibition of bacterial DNA gyrase and topoisomerase IV. SMX is a bacteriostatic antibiotic that primarily inhibits bacterial growth and reproduction by acting on folic acid synthesis [24]. Several studies evaluated the effects of CIP [25,26,27] and SMX on AD performance [28,29] and its microbial community. Although this research is of great interest, usually more than one antibiotic is detected in manure or digestate [17]. Studies investigating the synergistic effects of antibiotic mixtures on the AD process [30] and its microbial community are not so common [1,31] and, due to their complexity, are still incomplete. In addition, the roles that different microbial functional groups (hydrolytic-acidogenic-acetogenic-methanogenic) play in antibiotic biodegradation are still to be clarified [32].

In this study, the effects on the AD process of antibiotic mixtures of CIP and SMX were investigated using cattle manure as a substrate. Cumulative H_2_ and CH_4_ production, AD intermediates (i.e., volatile fatty acids—VFAs) and ethanol as well as antibiotic removal were considered with respect to the characteristics of the microbial community. The results obtained are compared to those of a previous study [33], carried out in the same experimental conditions and testing the effects of SMX alone, and discussed.

## 2. Results and Discussion

The results provide interesting information on the effects of the antibiotic CIP alone and in a mixture with SMX on the performance of the AD process, showing that the anaerobic microbial community is able to withstand the presence of the antibiotics and partially remove them.

### 2.1. Biogas Production and Process Intermediates

Under all experimental conditions, both CH_4_ and H_2_ were detected and the trends of their cumulative productions were observed (Figure 1).

In all cases, H_2_ production started immediately, reaching the maximum value at days 6–8. The start of CH_4_ production corresponded to the H_2_ decrease in the biogas on the timelines (Figure 1). The cumulative amount of H_2_ detected was comparable to that of CH_4_ production, for all experimental conditions (values ranging from 753.7 ± 102.4 mL L^−1^ to 2040.3 ± 20.2 mL L^−1^ detected for control and MIX_5, respectively). The control condition showed the lowest cumulative values of both H_2_ (*p* < 0.01) and CH_4_ (*p* < 0.001) and was also the only case showing twice as much H_2_ as CH_4_ (899.5 ± 101.1 mL L^−1^ and 386.8 ± 65.4 mL L^−1^, respectively).

The highest CH_4_ cumulative production (*p* < 0.01) was detected in MIX_2.5 with a value more than six-fold higher (2.742 ± 369 mL L^−1^) than the control. No significant differences were found at the end of the experiment among the other three spiked conditions, i.e., CIP_5, MIX_5 and MIX_10 (1.985 ± 228 mL L^−1^, 1.680 ± 148 mL L^−1^ and 1.919 ± 136 mL L^−1^, respectively). Despite this, the condition with only one antibiotic, i.e., CIP_5, reached maximum CH_4_ production already on day 40 of the experiment, whereas MIX_5 and MIX_10 needed more time, achieving the maximum cumulative production at days 68 and 83, respectively. Interestingly, among the conditions with the same amount of added antibiotics, i.e., MIX_2.5 and CIP_5, the latter, containing only ciprofloxacin, produced less CH_4_, revealing a detrimental effect if compared to its action in combination with SMX. A study carried out in the same experimental condition and spiking SMX alone at a concentration of 5 mg L^−1^ [33], achieved a final cumulative CH_4_ production of 2030.6 ± 143.3 mL L^−1^, a value comparable with those obtained in this experiment (excluding control and MIX_2.5 tests). Nevertheless, the authors observed that CH_4_ production in the SMX spiked test ended after only 33 days. This result confirms a lower impact of SMX on the AD microbial community than CIP, also due to its higher degradability. Other interesting information was obtained from the results on the composition (%) of the biogas at the end of the experiment (Appendix A). The control condition showed the lowest CH_4_ content in the biogas, with a concentration never exceeding 31%. Conversely, both CIP_5 and MIX_5 showed more than doubled CH_4_ concentrations, 73.1 ± 8.4% and 68.8 ± 6.4%, respectively, although these values were reached at days 35 and 69, respectively. In MIX_10 the CH_4_ content was 58.5 ± 6.4% (day 69).

A wide range of effects from complete inhibition to stimulation of CH_4_ production, depending on the antibiotic types and concentrations, was previously observed by other authors [7,9,27]. The aforementioned study by Mazzurco Miritana and colleagues [33] showed that the addition of SMX (5 mg L^−1^) to cattle manure resulted in higher CH_4_ production compared to the control condition. A similar result was reported by Zhi and Zhang [27] who observed that the addition of 100 mg L^−1^ of CIP to cattle manure stimulated CH_4_ production. Even in this study, the addition of antibiotics stimulated CH_4,_ as well as H_2_ production, highlighting that these results need further investigation to be fully understood. In particular, it needs to be verified whether antibiotics, in this case CIP, exerted selective pressure on specific functional components of the microbial community, probably competing with methanogens for the use of H_2_ (i.e., bacteria performing sulphate reduction and/or homoacetogenesis).

From this study emerges that the mixed conditions had the highest H_2_ production, and the latter hindered the AD process by delaying the start of CH_4_ production. Typically, the H_2_ produced during the AD process is not detected in significant concentrations in the biogas (i.e., ~5%) since it is mainly consumed by the microbial community for the final production of CH_4_. In this study, the high detection of H_2_ proved to be an indicator of the imbalance of the different phases of the AD process as widely discussed in Mazzurco Miritana et al. [33]. In particular, the H_2_ produced by *Bacteria* during the hydrolytic/acidogenic and acetogenic AD phases was not used by *Archaea* to produce CH_4_ as quickly as it was produced. Some studies reported that adding antibiotics inhibited acetate, propionate, and butyrate uptake [30,34]. Another study [35] reported possible effects of fluoroquinolones and sulfonamides on the *Archaea* domain affecting microbial growth and activity. In this study, the imbalance of the metabolic phases of the AD resulted in a process configured in two phases, with an earlier H_2_ production followed by a CH_4_ production phase. A similar condition was already observed in a previous study [36] in which imbalances between different functional groups in the microbial community resulted in imbalances in the AD process. The interpretation of the process imbalance is also supported by the results obtained from the analysis of the VFAs, lactic acid and ethanol, which were intermediates of the process (Figure 2a–e).

Although the ingestate collected already contained high concentrations of acetic (3.7 g L^−1^), propionic (2.5 g L^−1^) and lactic acids (1.6 g L^−1^) as well as ethanol (4.3 g L^−1^), in all cases the H_2_ production started immediately at the beginning of the experiment. Indeed, from the first two experimental weeks, a very high accumulation of acetic acid was observed in all experimental conditions, including the control (Figure 2a). The values even above 10,000 mg L^−1^ indicated the inability of the microbial community to promptly use it for producing methane. A previous study [4] reported that elevated hydrogen levels could temporarily inhibit the AD process by promoting the accumulation of acetate. In all spiked conditions, succinic, formic, valeric and butyric acids as well as ethanol were detected at values of a quarter and a fifth of the acetic acid (around 2000 mg L^−1^), indicating they were partially consumed during the acetogenic metabolic step (Figure 2b–e). An exception was represented by MIX_10, in which propionic acid persisted from day 9, with values around 6000 mg L^−1^. In the control condition, the acid concentrations were higher than in the spiked tests, particularly for butyric, propionic and valeric ones. The detection of increasing concentrations of acetic acid revealed that the rapid production of acids, as well as their transformation into acetic acid, was not followed by their use for methane production through the acetoclastic pathway, highlighting that there was a bottleneck in the AD process between the acetogenic and methanogenic metabolic phases. In contrast, hydrogen consumption, in line with the start of CH_4_ production, clearly showed that methanogenesis was performed by the hydrogenotrophic metabolic pathway.

### 2.2. Antibiotic Removal

In the course of the experiment, CIP and SMX were monitored, and the efficiency of their removal was quantified. The results obtained (Figure 3) show that both CIP and SMX were degraded during the AD process. On the other hand, the antibiotic mixtures and increasing concentrations hindered their complete depletion.

CIP alone (i.e., CIP_5) degraded three times less when combined with SMX in MIX_5, showing 20.9% and 6.5% removal, respectively. This difference, observed in tests conducted with the same final antibiotic concentration, is attributable to the synergism generated by the different mechanisms of action exerted by the two antibiotics investigated on microorganisms. The lowest and highest concentrations of antibiotic mixtures, i.e., MIX_2.5 and MIX_10, showed the highest (8.1%) and lowest (5.6%) CIP removal, respectively. In any case, CIP removal never exceeded 21%, confirming this antibiotic to be a recalcitrant compound in natural and artificial ecosystems, as reported by other authors [15,18,25]. SMX was degraded more easily than CIP. A percentage of 49.0% of its removal was reached in MIX_2.5, but its degradation was also strongly negatively influenced by the co-presence of CIP. Indeed, Mazzurco Miritana et al. [33] observed that SMX alone was removed 81% at the end of their experiment. Some aspects remain to be clarified concerning the role of SMX in the mixtures. Indeed, at the end of the experiment, Mix_10 produced more CH_4_ than Mix_5 (Figure 1), in line with the removal rates (Figure 3). On the other hand, SMX_5 produced CH_4_ faster than CIP_5, since the production plateau was reached at days 22 (SMX_5) and 40 (CIP_5), showing a lower effect on the microbial community of SMX than that of CIP.

It is very interesting to note that, in all the experimental conditions, antibiotic removal only occurred until day 21, during H_2_ production, corresponding to the hydrolytic/acidogenic metabolic step of AD. Indeed, the maximum H_2_ peaks were reached within the first 21 days of the experiment, suggesting that the antibiotic removal was therefore ascribable to the hydrolytic-fermenting microorganisms able to break down complex molecules into simpler ones. On the contrary, Mazzurco Miritana et al. [33] observed that SMX alone degraded even after day 21, and at the end of the experiment (69 days) there was just 20% of its initial concentration.

### 2.3. Microbial Community Analysis

The values of the total microbial abundance were in line with the H_2_ and CH_4_ production trends. Under all spiked conditions, an initial detrimental effect of antibiotics on microbial abundance was found, as compared to the control (Figure 4a), demonstrating that antibiotics negatively affected microbial populations within the reactors. However, this effect was transient and an increase in microbial abundance was observed in the spiked conditions. The mixtures MIX_2.5, MIX_5 and MIX_10 showed the highest microbial abundances in the first 15 experimental days (days 15 and 6, respectively), in line with their maximum H_2_ production rate (Figure 1). In particular, MIX_2.5, the second one in cumulative H_2_ production, and the first one in cumulative CH_4_ production, showed the highest microbial abundances, at days 6 and 15 (6.2 × 10^9^ ± 4.6 × 10^8^ cells mL^−1^ and 7.91 × 10^9^ ± 8.35 × 10^8^ cells mL^−1^, respectively), while MIX_5 reached its maximum microbial abundance at day 6 (7.5 × 10^9^ ± 6.62 × 10^8^ cells mL^−1^), in line with the highest H_2_ production detected during the experiment. Conversely, the control condition showed the highest microbial abundance in the first experimental week and then gradually decreased. At the end of CH_4_ production, microbial abundance values dropped in all experimental conditions.

The FISH analysis clearly showed that adding antibiotics always had detrimental effects on archaeal abundance (N. cells mL^−1^) (Figure 4b) as well as on *Archaea* percentages (%) (Figure 4c), in line with antibiotic concentrations.

A number of previous studies reported the effects of antibiotics on the *Bacteria* and *Archaea* domains [26,27,37], showing that macrolides inhibited more methanogens than *Bacteria*, in particular acetoclastic. Moreover, it must be considered that the *Archaea* component of the AD microbial community thrives by relying on *Bacteria* metabolism. The antibiotic impact on specific functional bacterial components could also therefore result in the indirect and unexpected elimination of antibiotic-resistant *Archaea* [38]. This study shows that the highest CH_4_ production efficiency observed under spiked conditions should not be sought solely in the composition and abundance of microbial components, but further studies are needed to identify the AD parameters affecting microbial community functionality.

A few studies have discussed the proportion of fermentative *Bacteria versus* methanogenic *Archaea* [36,39], although it could prove to be a key factor in assessing the efficiency of the AD process. To our knowledge, this study reported the proportion of *Bacteria*/*Archaea* cells under different experimental conditions and at different experimental times for the first time (Table 1).

It is made evident that the control condition showed few variations during the experiment while the spiked conditions, particularly the MIXs, showed the highest values, thus demonstrating an increased proportion of *Bacteria*. A similar result was reported by Mustapha et al. [39]. The authors reported that azithromycin in sludge promoted an increased proportion of *Bacteria*, corresponding, in this case, to an increase in hydrolysis efficiency. In turn, a high bacterial concentration can lead to a further accumulation of the intermediate VFAs (if they are not readily converted to methane), as observed also in this study. Moreover, although *Archaea* can be resistant to antibiotics, some metabolic groups appear to be more susceptible [38]. A previous study reported that hydrogenotrophic methanogens had low sensitivity to CIP and SMX antibiotics [26].

The microbial community analysis performed using the MiSeq Illumina platform (Figure 5a,b) showed initial percentages of 99% *Bacteria* and <1% *Archaea*, respectively (Figure 5b). *Methanobrevibacter* and *Methanosphaera* were the dominant genera in the *Archaea* guild (Figure 5a), with the latter dominant at the end of the experiment in the MIXs conditions.

Both genera produce methane through the hydrogenotrophic pathway. In particular, *Methanosphera* obtain energy for growth by using hydrogen to reduce methanol to methane [40]. The results confirm that hydrogenotrophic methanogens predominated in the microbial community, as was supposed on the basis of the results obtained from the analysis of process intermediates and trends in H_2_ content in the biogas. Only a small fraction of acetoclastic genera, *Methanosarcina* and *Methanosaeta,* were detected.

An analysis of the microbial community at class level (Figure 5b) showed that *Clostridia* made up more than half of the community and only declined under mixed conditions at the end of the experiment. *Actinobacteria*, *Bacilli*, *Bacteroidia* and *Gammaproteobacteria* were the other main components of the community.

The principal coordinate analysis based on Bray–Curtis distances (Figure 6) showed significant differences (Permanova, *p* < 0.001) among the conditions considering the prokaryotic communities.

In particular, the controls were different from the MIX conditions at the end of the experiment and of the other conditions (i.e., MIX conditions at 15 and 21 days and the CIP_5 at 15 and 21 days). The CIP_5 at day 21 was in the middle, between the control group and the other condition group.

The diversity indices are reported in Table 2. The Chao1 and Shannon (H) indices in antibiotics treated samples were lower than in the control at all sampling times.

This decrease in diversity can be associated with the detrimental effect of the antibiotics on the prokaryotic community. Among the antibiotic-treated conditions, the MIXs at 15 and 21 days showed microbial diversity values lower than CIP_5 (Table 2). This result can be ascribed to a stronger combined effect of the antibiotic mixtures on some microbial populations. However, the decrease in microbial diversity did not affect the community functionality since antibiotic spiked conditions produced higher amounts of CH_4_ than the control, showing the presence of resistant prokaryotes. In particular, it can be stated that antibiotics exerted a selective effect on methanogenic *Archaea* favoring the genera of *Methanobrevibacter* and *Methanosphera*, which were very effective in the production of methane along the hydrogenotrophic metabolic pathway.

## 3. Materials and Methods

### 3.1. Anaerobic Digestion Test

The experimental setup of the AD tests is detailed in a previous work [33]. Briefly, the feeding ingestate of a full-scale AD plant, consisting of fresh cattle manure, was used to perform batch AD tests spiked with antibiotics. In particular, the ingestate was collected from the feeding tank pipe of a full-scale reactor located in a beef and dairy cattle farm in Central Italy (Lazio region). The tank collected the cattle manure produced daily and, using a pump, sent it to the reactor. Aliquots were collected by filling glass bottles of 2 L each (three replicates) after the pump of the collecting tank had been running for 15 min. The bottles were transported to the laboratory where the ingestate was analyzed to detect possible residual amounts of SMX (0.3 mg L^−1^) and CIP (0.1 mg L^−1^) before setting up the experiment. The bottles were transported to the laboratory where the experiment was immediately set up. The ingestate used was characterized as reported in a previous work [33]. Five experimental conditions (3 replicates each) were set up by adding antibiotics as follows: CIP alone (5 mg L^−1^), three mixtures of CIP and SMX (1:1 ratio) at concentrations of 2.5, 5, 10 mg L^−1^ of each antibiotic (hereinafter referred as CIP_5, MIX_2.5, MIX_5, MIX_10, respectively) and a control condition, set up without antibiotic spiking. The antibiotic concentrations were established by taking into account our previous study evaluating the presence of SMX, CIP and ENR in a full-scale anaerobic plant [17] located in the same geographical area.

Antibiotic single stock solutions were prepared by dissolving powdered CIP (purity 99%, Sigma-Aldrich) and SMX (purity 99%, Sigma-Aldrich) in a methanol/MilliQ solvent. The solutions obtained were further diluted in ultrapure water to reach the final concentrations to be tested.

The AD tests were set up using 600 mL Pyrex glass bottles filled with 300 mL of ingestate corresponding to 31 g volatile solids (VS). The bottles were sealed with a rubber stopper and metal ring and flushed with N_2_ (10 s) to establish anaerobic conditions. Antibiotic solutions were added by using syringes and each bottle was shaken for 10 s. A solution of ultrapure water and methanol was added to the control conditions. The final amount of methanol in each batch was 0.18 mL. The samples were immediately collected in order to check the concentration of antibiotics at the beginning of the experiment. Throughout the experiment, the reactors were kept at a constant temperature of 37 to 38 °C. Measurements of biogas production and its CH_4_ content, as well as liquid medium sampling, were performed daily during the first 6 experimental days. Subsequently, samples were collected once a week until day 27, then once every fortnight until the end of biogas production. Each experimental test was considered completed when no more CH_4_ was produced for two weeks [36].

### 3.2. Biogas and Organic Acid Measurements

Biogas volumetric production was measured using a water displacement device [41] and H_2_, CH_4_ and CO_2_ content (%) in the biogas was analyzed using a gas chromatograph (Focus GC, by Thermo Scientific, Waltham, MA, USA) equipped with a thermal conductivity detector (TCD) and a 3 m stainless-steel column packed with Hayesep Q (800/100 mesh). Nitrogen gas was used as a carrier at a flow rate of 35 mL min^−1^. The temperature of the column and of the injector was 120 °C, while that of the TCD was 200 °C.

Cumulative H_2_ and CH_4_ productions were calculated with Logan equation [42]. The composition of process intermediates in terms of acetic, succinic, lactic, formic, propionic, butyric and valeric acids, as well as ethanol, was analyzed using High Performance Liquid Chromatograph (HPLC) technology. Liquid samples were diluted 1:10 in H_2_SO_4_ 5 mN and filtered with a 0.22 µm membrane before injection into the HPLC. A Thermo Spectra system (USA), equipped with a UV detector (λ 210 nm) and 300 mm × 7.8 mm Rezex ROA-Organic Acid Hþ (8%) column (Phenomenex, Torrance, CA, USA) with a 4 × 30 mm Carbo-H security guard cartridge (Phenomenex, USA), was operated at 75 °C, by using a 5 mN H_2_SO_4_ solution as the mobile phase (flow rate, 0.5 mL min^−1^).

### 3.3. Antibiotic Determination 

The analytical determination of SMX and CIP were performed as reported by Visca et al. [17]. Briefly, PLE (Pressurized Liquid Extraction, E-916 Speed Extractor, Büchi, Italy) allowed the extraction of the target antibiotics from 2 g samples using a mixture of MeOH/ACN (1:1, *v*/*v*) as solvent. The extracts were then cleaned-up/purified by SPE (Solid Phase Extraction, Supelco, Bellefonte, PA, USA) equipped with Oasis HLB cartridges properly activated. The SMX and CIP concentrations were determined using HPLC-MS/MS (HPLC-column Oven mod. LC-100 and micro–Pump Series 200, Perkin Elmer, MA, USA; MS/MS, API 3000, AB Sciex, Darmstadt, Germany) fitted with an electrospray ionization source. The chromatographic column was Gemini (150 × 4.6 mm, 5 µm RP C 18, Phenomenex, Sartrouville, France). The injection volume and the flow rate were 20 µL and 0.3 mL min^−1^, respectively. The mobile phase was composed of MeOH (phase A) and an aqueous formic acid solution (0.1%) (phase B). The chromatographic run and the main MS/MS parameters are detailed by Visca et al. [17]. The HPLC-MS/MS system and data acquisition were controlled by the Analyst^®^ 1.6 Software (AB Sciex, Concord, ON, Canada). 

A good linearity was obtained for the concentration range of 0.5–7.5 mg L^−1^ for the two antibiotics, as indicated by the correlation coefficient (R^2^, always >0.99). Calibration standards (in the range of 0.1–7.5 mg L^−1^) were prepared in triplicate for three validation runs performed on different days. The relative standard deviations of the concentration tested were <15%. The internal standard calibration was performed with the addition of deuterated standards (sulfamethoxazole-d4 and ciprofloxacin-d8 hydrochloride hydrate) to both working standard solutions and purified extracts.

Recovery was evaluated by spiking feeding ingestate matrix with the target antibiotics alone and in a mixture (1:1 ratio) at three different concentrations (1, 5 and 10 mg L^−1^, five replicates). The average recovery rates for SMX and CIP were 107.4 ± 6%, and 68.2 ± 5%, respectively. No reduction in the average recoveries was measured when antibiotics were spiked as a mixture.

The limits of detection (LOD) for the SMX and CIP antibiotics [43] were 1.5 µg L^−1^ and 2.0 µg L^−1^, respectively. The quantification limits were set as three times LOD values.

### 3.4. Microbiological Analysis 

Liquid medium samples (triplicates) were fixed as described in Pernthaler et al. [44] and stored (−20 °C) until use. The timing for microbiological investigations was chosen in line with the performance of the AD process. Total microbial abundance was determined at days 3, 6, 10, 15, 21, 27 and at the end of CH_4_ production by using direct cell count (N. cells mL^−1^) after staining with DAPI (4′,6-diamidino-2-phenylindole, 1 µg mL^−1^ each sample, 3 replicates, 20′). The samples were then collected on black polycarbonate filters (pore size 0.22 μm, diameter 25 mm, Millipore, Burlington, MA, USA). Filters were placed on slides and examined (20 fields) using a Zeiss epifluorescence microscope AXIOSKOP 40 (Carl Zeiss, Jena, Germany) equipped with a ZEISS HXP 120v Light Source (1000× magnification). Microbial community structure was analyzed using the fluorescence in situ hybridization (FISH) technique as described in Amann et al. [45] and Barra Caracciolo et al. [46] at days 3, 6, 10, 15, 21 and at the end of CH_4_ production. Before analysis, a cell extraction procedure was performed to detach and separate microbial cells from inorganic particles as described in Barra Caracciolo et al. [47]. Each sample (3 replicates) was collected on white polycarbonate filters (pore size 0.22 μm, diameter 25 mm, Millipore, Burlington, MA, USA). The EUB338, II, III and ARC915 probes were used for the detection of *Bacteria* and *Archaea* active cells [48]. The cell average enumeration was performed in order to evaluate the percentage of positive signals versus DAPI-stained positive cells (20 fields for each slide).

### 3.5. DNA Extraction and Next Generation Sequence (NGS)

The genome of the prokaryotic community was sequenced with a next-generation sequencing (NGS) using the MiSeq platform (Illumina). The Pro341F and Pro805R primers were used to amplify the V3-V4 region of 16S rRNA genes. These primers were selected in order to ensure the simultaneous identification of *Bacteria* and *Archaea* [49]. FastQ files were imported using QIIME2 v2019.11 [50] and denoised with the DADA2 plugin [51]. The primers were removed using the “trim-left-f” (forward) and “trim-left-r” (reverse) DADA2 functions. The length of the primers was 17 nucleotides for the forward and 21 nucleotides for the reverse one. The amplicon sequencing variants (ASVs) with less than 0.005% of high-quality reads were then filtered and compared to the 97% identical clustered Ribosomal Database Project (RDP release 11) using a naive Bayes classifier trained on the amplified region with an 80% confidence. The bacterial diversity was analyzed using the Evenness and Shannon diversity indices, while the Chao 1 index [52] was used as an estimator of potential richness (calculated with Qiime2 using the *diversity alpha* function).

### 3.6. Statistical Analysis

The Kruskal-Wallis test (a non-parametric one-way ANOVA) was used to evaluate the differences within the alpha diversity indices and was calculated by R (4.0.4 version https://www.r-project.org/ (accessed on 15 June 2022)), using the *kruskal.test* function together with the *pairwise.wilcox.test* function, as the post-hoc test [53].

The principal coordinate analysis based on the Bray-Curtis distance and PERMANOVA as the statistical test was performed using the online tool Microbiome Analyst (http://www.microbiomeanalyst.ca (accessed on 15 June 2022)) in order to evaluate the composition of the bacterial community during the AD process.

Finally, all the histograms and stacked bar plots were constructed with MS Excel 2013.

## 4. Conclusions

Adding SMX and CIP alone or in mixture did not negatively affect the AD process, showing that the functional prokaryotic populations involved in this process were previously adapted to antibiotic effects. To confirm this, the cumulative production of CH_4_ was enhanced. This means that antibiotic-contaminated manure non used for AD, and left exposed to air, can produce more methane in the anaerobic inner part of the piles, increasing climate-altering gas emissions. Moreover, because SMX and CIP decreased substantially in digestate, the latter is more desirable as an organic fertilizer in order to reduce the agricultural soils antibiotic contamination and presumably antibiotic resistant genes.

Finally, the antibiotics influenced the structure of the microbial community since *Bacteria* increased and *Archaea* decreased with the rise in antibiotic concentrations. In particular, at the end of the experiment, a predominance of archaeal genera (*Methanobrevibacter* and *Methanosphera*), effective in the production of methane along the hydrogenotrophic metabolic pathway, was observed. In contrast, the presence of antibiotics counteracted the functionality of acetotrophic methanogens, as confirmed by the acetic acid increased concentrations detected until the end of the experiment.

## Figures and Tables

**Figure 1 antibiotics-11-01111-f001:**
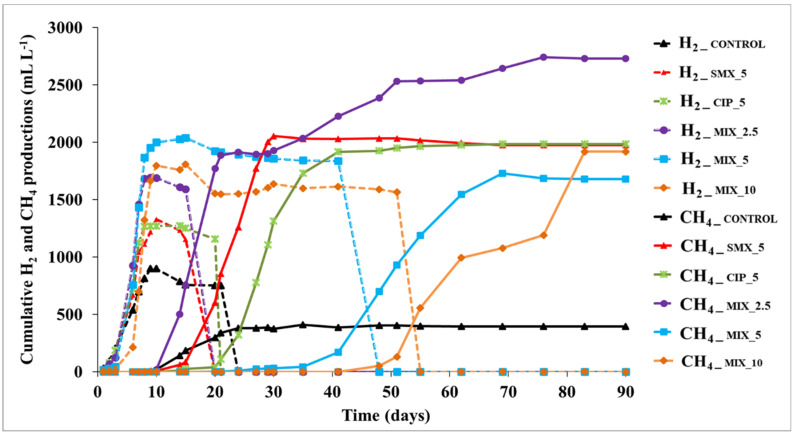
Cumulative H_2_ and CH_4_ production in all experimental conditions. Dotted lines and solid lines represent the cumulative production of H_2_ and CH_4_, respectively (standard deviations < 10%). The values of the SMX_5 test are those reported in Mazzurco Miritana et al. [33].

**Figure 2 antibiotics-11-01111-f002:**
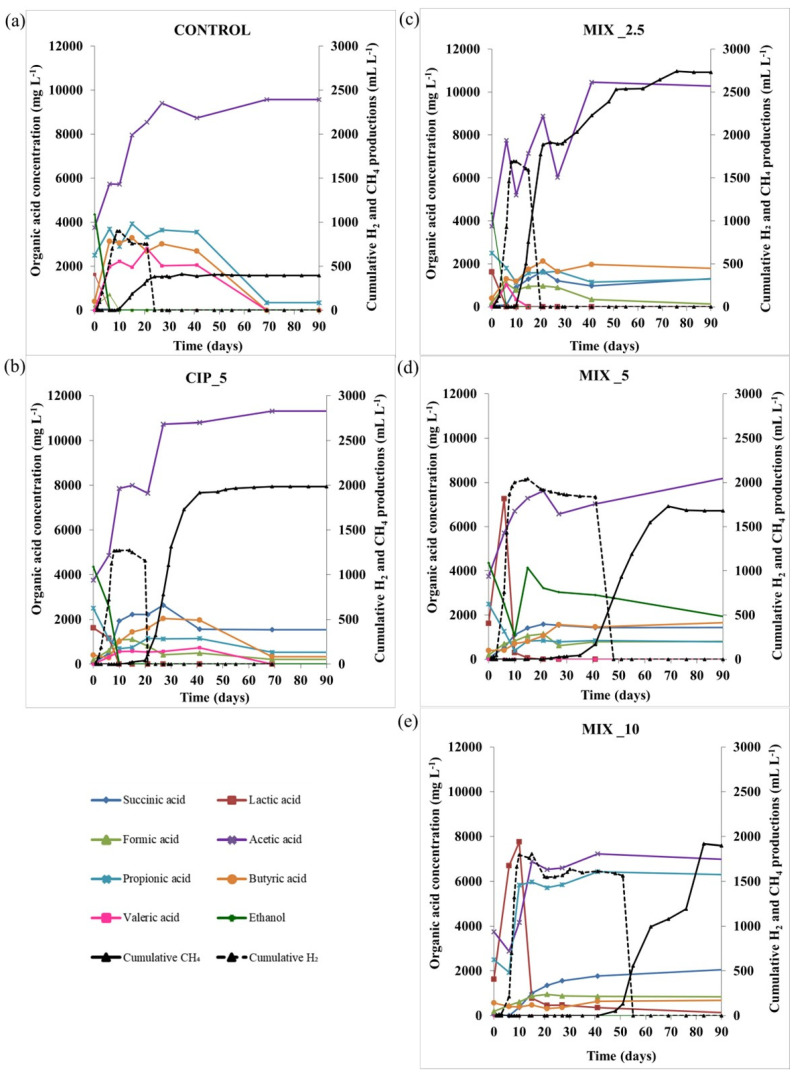
Organic acids and ethanol concentrations in the different conditions (**a**–**e**) during the anaerobic digestion process compared with H_2_ and CH_4_ production (black dotted and solid lines represent H_2_ and CH_4_ production, respectively). The standard deviations are <10%.

**Figure 3 antibiotics-11-01111-f003:**
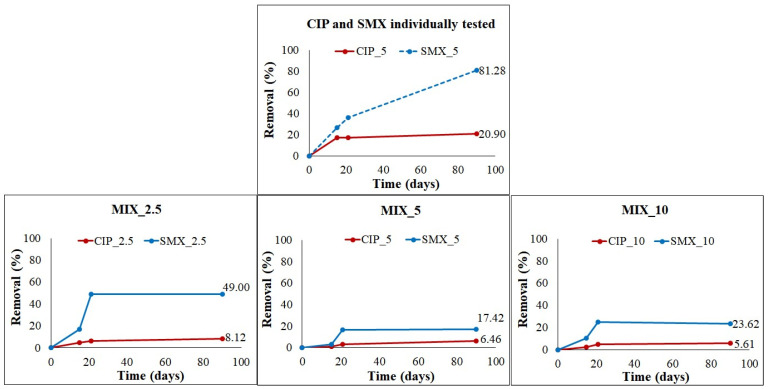
Antibiotics removal (%). Dotted line represents the results already published in a previous study [33].

**Figure 4 antibiotics-11-01111-f004:**
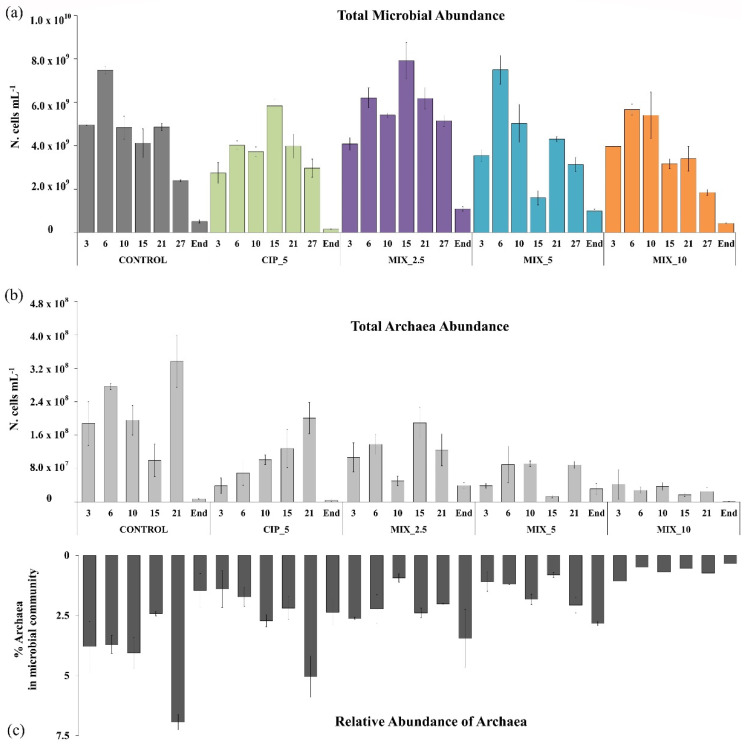
(**a**) Total microbial abundance (N. cells mL^−1^) detected by direct counts after DAPI staining for all experimental conditions over the experimental time (days 3, 6, 10, 15, 21, 27, end). (**b**) Microbial abundance (N. cells mL^−1^) of *Archaea* estimated combining DAPI and FISH results. (**c**) Relative abundance (%) of *Archaea*.

**Figure 5 antibiotics-11-01111-f005:**
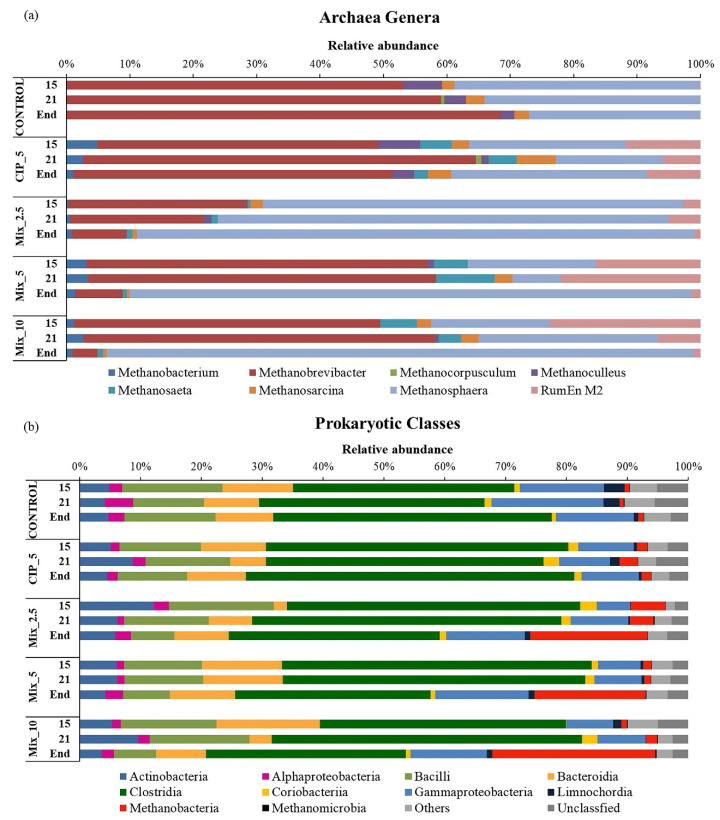
(**a**) Relative abundance of *Archaea* genera in all experimental conditions evaluated by NGS at genus level and (**b**) relative abundance of prokaryotic cells in all experimental conditions evaluated by NGS at class level.

**Figure 6 antibiotics-11-01111-f006:**
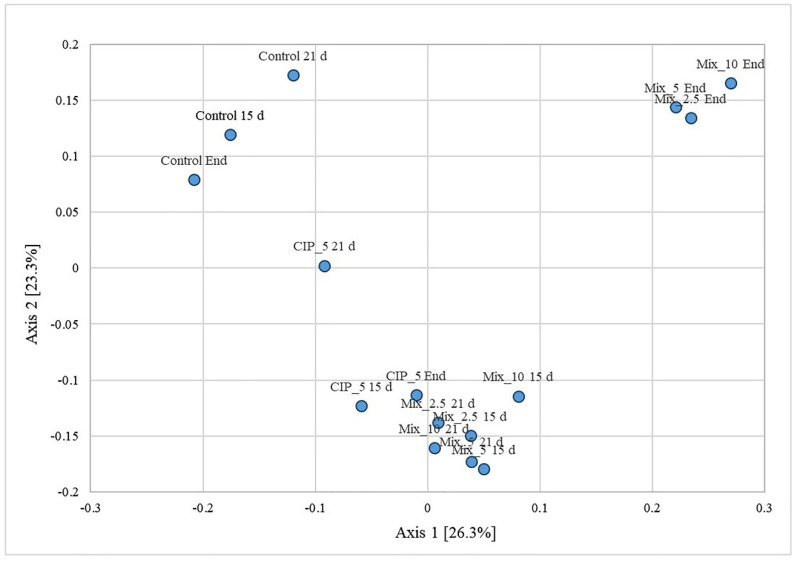
Principal Coordinate Analysis based on Bray–Curtis. The end-time is day 69.

**Table 1 antibiotics-11-01111-t001:** Ratio of bacterial/archaeal cells under different experimental conditions and at different experimental times, calculated from the microbial abundance values of the respective components.

	*Bacteria/Archaea*				
Time (Day)	Experimental Conditions				
	**Control**	**CIP_5**	**Mix_2.5**	**Mix_5**	**Mix_10**
3	18.5	57.7	27.0	63.6	66.4
6	19.4	36.9	33.8	60.8	141.4
10	18.4	27.6	70.6	39.6	102.2
15	30.0	31.0	30.7	86.4	133.7
21	9.6	16.0	34.6	37.1	76.8
end	38.8	30.0	17.5	23.6	180.8

**Table 2 antibiotics-11-01111-t002:** Shannon Weaver diversity index (H’), Pielou’s evenness index (E) and Chao index calculated for all experimental conditions at days 15, 21 and at the end of the experiment.

Time	Index	Experimental Conditions				
		Control	CIP_5	Mix_2.5	Mix_5	Mix_10
day 15	H’	7.13	6.25	6.05	6.13	6.06
	E	0.76	0.71	0.73	0.72	0.72
	Chao1	358.95	281.60	223.17	259.10	233.50
day 21	H’	6.84	6.47	6.18	6.17	6.06
	E	0.75	0.75	0.73	0.72	0.72
	Chao1	324.80	264.50	265.83	265.15	251.55
end	H’	6.86	6.24	6.17	6.23	5.88
	E	0.76	0.70	0.70	0.71	0.66
	Chao1	322.41	287.36	282.19	279.00	287.53

## Data Availability

Not applicable.

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
