# Peer review of "Effects of Ciprofloxacin Alone or in Mixture with Sulfamethoxazole on the Efficiency of Anaerobic Digestion and Its Microbial Community"

_antibiotics, 2022, doi:10.3390/antibiotics11081111_

Round 1
Reviewer 1 Report
Mazzurco Miritana et al. explored the effects of CIP and SMX mixed together on the anaerobic digestion of microbial communities. The study is for the most part well written. The experiments and results are mostly well presented and I agree with the overall conclusions. There are a few pieces in this manuscript that should be revised and improved. I am breaking this down as follows:
Abstract: define CIP here, as many of the readers may be confused.
Introduction: This already contains a fair amount of language pointing out the main rationale to conduct this study, but I believe this could rephrased and explained a little more clearly. What is the problem right now with the AD in livestock manure? How is this study directly addressing this problem, or at least parts of it.
Results and Discussion:
line 133: "the higher the H2 production" I believe you mean "the longer (in days) the H2 production".
line 143 to 145: This statement seems to refer to this study and not citation [36]. Is there a need to put citation [36]?
line 196 to 199: As I read the paper, at this point in the study there is only a correlation, but no direct evidence to support this claim.
line 209 to 211: MIX_2.5 peeked at 15; MIX_5 peeked at day 6; MIX_10 peeked at day 6. Either I am reading the figure wrong or something else is wrong.
line 221: (N. cells mL-1) no need to put the L.
Figure 4a: The colors of the bars are going across conditions. Please make each condition one color. CIP_5 and MIX_5 are missing day 3. Is that correct? if so, please provide an explanation.
Figure 4c: Please rephrase the Y axis label to "% Archea in microbial community". Also, does it have to have the bars facing down? Or is it that you are trying to use the X axis labels from 4b on 4c as well? It is fine either way. CONTROLLO would be CONTROL, although most people would understand the Italian word.
line 227: Data on SMX alone are also reported? I couldn't find it in this manuscript.
line 256: please define VFAs
Figure 5a and 5b show that Methanosphera are clearly increasing with the treatments. In addition, there is an evident increase in the methanogenic bacteria with treatments as well. Would you consider discussing the possibility that these bacteria could also contribute to higher CH4 production? There is language about CH4 production in lines 298 - 303. Perhaps this point may fit there?
Methods:
line 309: "The ingestate was collected from the feeding tank pipe of a full-scale reactor..." Does this mean that this material was not inside the animal's gut? Or is this fresh cow manure that is being collected to a reactor? I am a bit confused because of the word "ingestate" which sounds to me like the animal just had a meal and it is digesting this food in the GI tract. Perhaps it would be helpful to define "ingestate" in the introduction to reduce any confusion.
Most everything else is already in great shape. All the best with the publication of this study.
Author Response
ANSWERS TO REVIEWER OBSERVATIONS
First of all, we would like to express our sincere acknowledgements to the anonymous Reviewer for
the very good suggestions for improving our manuscript.
We have corrected all the manuscript in accordance with the comments (see below and in blue
colour in the text).
Reviewer 1
Mazzurco Miritana et al. explored the effects of CIP and SMX mixed together on the anaerobic
digestion of microbial communities. The study is for the most part well written. The experiments
and results are mostly well presented and I agree with the overall conclusions. There are a few
pieces in this manuscript that should be revised and improved. I am breaking this down as follows:
Abstract: define CIP here, as many of the readers may be confused.
Answer: Thank you for your observation. We have defined it in the text.
Introduction: This already contains a fair amount of language pointing out the main rationale to
conduct this study, but I believe this could rephrased and explained a little more clearly. What is
the problem right now with the AD in livestock manure? How is this study directly addressing this
problem, or at least parts of it.
Thank you for your useful suggestion. Livestock manure can contain antibiotic residues and using it
as an organic fertilizer can introduce these emerging contaminants in agricultural soils. The paper
addresses the effect of antibiotics on the AD process, evaluating the biogas production and the
possible antibiotic decrease in the digestate.
We have specified better “the problem” with the livestock manure in the introduction, see lines 46-
48 and 71-74. Moreover, in the conclusion we have specified the advantages of using digestate as
an organic fertilizer (lines 449-456).
Results and Discussion:
line 133: "the higher the H2 production" I believe you mean "the longer (in days) the H2
production".
Thank you for this observation. We have clarified the sentence by writing “From this study, it
emerges that mixed conditions have the highest H2 production and the latter hinders the AD process
by delaying the start of CH4 production”.
In fact, H2 production occurs in the first 10 experimental days but the Archaea struggles to utilise it
for CH4 production and the result is that the H2 concentration remains stationary for many days
(even up to 55 days in the case MIX_10). Lines 134-135
line 143 to 145: This statement seems to refer to this study and not citation [36]. Is there a need to
put citation [36]?
Thank you for this observation. We added the sentence “A similar condition was already observed
in a previous study [36] in which imbalances between different functional groups in the microbial
community resulted in imbalances in the AD process”
line 196 to 199: As I read the paper, at this point in the study there is only a correlation, but no
direct evidence to support this claim.
Thank you very much for your fitting observation Yes, it is indeed a deduction. In fact, we have
used the phrase "suggesting that the antibiotic removal is therefore attributable ....." (lines 207-210).
line 209 to 211: MIX_2.5 peeked at 15; MIX_5 peeked at day 6; MIX_10 peeked at day 6. Either I
am reading the figure wrong or something else is wrong.
Thank you very much. We have changed the sentences in: “The mixtures, MIX_2.5, MIX_5 and
MIX_10, showed the highest microbial abundance in the first 15 experimental days (days 15 and 6
respectively), in line with their maximum H 2 production rate (Figure 1)” (lines 220-222).
line 221: (N. cells mL-1) no need to put the L.
Thank you very much. Done
Figure 4a: The colors of the bars are going across conditions. Please make each condition one color.
CIP_5 and MIX_5 are missing day 3. Is that correct? if so, please provide an explanation.
Thank you very much. You are right. We have corrected the mistakes and we have replaced the
right figure into the text.
Figure 4c: Please rephrase the Y axis label to "% Archaea in microbial community". Also, does it
have to have the bars facing down? Or is it that you are trying to use the X axis labels from 4b on 4c
as well? It is fine either way. CONTROLLO would be CONTROL, although most people would
understand the Italian word.
Thank you for your observation. We have update the Figure 4c in accordance with your
suggestions. We have rephrased the Y axis label with "% Archaea in microbial community". We
have turned the bars down in order to use the labels of the graph 4b above. We have also corrected
the mistake for the CONTROL word, and we have update the Figure in accordance with your
suggestions
line 227: Data on SMX alone are also reported? I couldn't find it in this manuscript.
Answer: Sorry, you are right. The data on SMX alone (SMX_5) are not reported in this paper. They
are however available in our previous publication Mazzurco Miritana et al. 2020, (33)
line 256: please define VFAs
Thank you. In the text we have defined volatile fatty acids (VFA) the first time at line 73; then we
have also used the acronym at lines 150 and 266.
Figure 5a and 5b show that Methanosphera are clearly increasing with the treatments. In addition,
there is an evident increase in the methanogenic bacteria with treatments as well. Would you
consider discussing the possibility that these bacteria could also contribute to higher CH4
production? There is language about CH4 production in lines 298 - 303. Perhaps this point may fit
there?
Answer: Thank you for your interesting observation. We have improved the pre-existing sentence
and also added another one specifying the role of methanogenic Archaea:
“This decrease in diversity can be associated with the detrimental effect of the antibiotics on the
prokaryotic community. Among the antibiotic treated conditions, the MIXs at 15 and 21 days
showed microbial diversity values lower than CIP_5 (Table 2). This result can be ascribed to a
stronger combined effect of the antibiotic mixtures on some microbial populations. However, the
decrease in microbial diversity did not affect the community functionality since antibiotic spiked
conditions produced higher amounts of CH 4 than the Control, showing the presence of resistant
prokaryotes. In particular, it can be stated that antibiotics exerted a selective effect on methanogenic
Archaea favouring the genera of Methanobrevibacter and Methanosphera, which were very
effective in the production of methane along the hydrogenotrophic metabolic pathway” (lines 304-
313).
lines 298 – 303 In particular, it can be stated that antibiotics exert a selective effect on
methanogenic Archaea favouring the genera of Methanobrevibacter and Methanosphera, which are
very effective in the production of methane along the hydrogenotrophic metabolic pathway”
Methods:
line 309: "The ingestate was collected from the feeding tank pipe of a full-scale reactor..." Does this
mean that this material was not inside the animal's gut? Or is this fresh cow manure that is being
collected to a reactor? I am a bit confused because of the word "ingestate" which sounds to me like
the animal just had a meal and it is digesting this food in the GI tract. Perhaps it would be helpful to
define "ingestate" in the introduction to reduce any confusion.
Answer: Thank you for your useful observation. We have improved the description of the sampling
as reported “Briefly, the feeding ingestate of a full-scale AD plant, consisting of fresh cattle
manure, was used to perform batch AD tests spiked with antibiotics. More in particular, the
ingestate was collected from the feeding tank pipe of a full-scale reactor located in a beef and dairy
cattle farm in Central Italy (Lazio region). The tank collected the cattle manure daily produced and,
using a pump, sent it to the reactor” (lines 317-322).
Most everything else is already in great shape. All the best with the publication of this study.
Thank you very much for your encouragement, it is very helpful to us!

Reviewer 2 Report
Were the samples used for the AD setup antibiotic free? This should be mentioned as any present could effect the test samples (potential synergistic effects).
Figure 1 is very crowed and this makes it very difficult to see the values and understand what is going on. I would suggest dividing it into figure 1a and 1b splitting into CH4 and H2
In Figure 2 the Figure key is way too big and the actual figures way too small.
In line 312 what does the number 15 denote? Days? Weeks?
Author Response
Manuscript ID: antibiotics-1825556 by Mazzurco Miritana et al., entitled “Effects of ciprofloxacin
alone or in mixture with sulfamethoxazole on the efficiency of anaerobic digestion and its microbial
community" investigates the impact of antibiotics on microbial community functionality in
anaerobic digestion (AD) process of cattle manure.
General remarks:
The proposed manuscript is well written and provides new insight into the possible factors affecting
biogas production. Antibiotics are often used in excess in modern livestock farms but their
subsequent impact on other environmental processes is often overlooked. Here I must disagree with
the authors that “AD proved to be a valuable treatment for reducing residual concentrations of CIP
and SMX antibiotics from livestock manure”. Your results concerning the removal of antibiotics
(Figure 3) (on average 6% and 30% for CIP and SMX, respectively) as well as literature data
indicate that antibiotics have a strong impact on microbiomes. In my opinion, you must remove this
statement from conclusions and add at least a short discussion that antibiotics have a higher impact
on acetoclastic methanogens than hydrogenotrophic (reference 1 from below), and considering
hydrogenotrophic methanogens occurrence in environmental samples (2) the remaining antibiotics
may have a relatively low impact on natural biogas release but it needs to be highlighted that
antibiotics can impair acetate utilization and many other biological processes, promote the transfer
of antibiotic resistance genes to indigenous environmental bacteria and Archaea (3).
11. (9 from your reference list) Cheng, D.L.; Ngo, H.H.; Guo, W.S.; Chang, S.W.; Nguyen, D.D.;
Kumar, S.M.; Du, B.; Wei, Q.; Wei, D. Problematic effects of 487 antibiotics on anaerobic
treatment of swine wastewater. Bioresource Technology 2018, 263, 642–653, 488
doi:10.1016/J.BIORTECH.2018.05.010
22. Pyzik, A., Ciezkowska, M., Krawczyk, P. S., Sobczak, A., Drewniak, L., Dziembowski, A., &
Lipinski, L. (2018). Comparative analysis of deep sequenced methanogenic communities:
identification of microorganisms responsible for methane production. Microbial cell factories,
17(1), 1-16. https://doi.org/10.1186/s12934-018-1043-3
33 . Xu, R., Yang, Z. H., Zheng, Y., Wang, Q. P., Bai, Y., Liu, J. B., ... & Fan, C. Z. (2019).
Metagenomic analysis reveals the effects of long-term antibiotic pressure on sludge anaerobic
digestion and antimicrobial resistance risk. Bioresource technology, 282, 179-188.
https://doi.org/10.1016/j.biortech.2019.02.120
Thank you for your useful suggestions: We have re-written the conclusions as follows:
“Adding SMX and CIP alone or in mixture did not affect negatively the AD process, showing that
the functional prokaryotic populations involved in this process were previously adapted to antibiotic
effects. To confirm this, the cumulative production of CH4 was enhanced. This means that
antibiotic-contaminated manure non used for AD, and left exposed to air, can produce in the
anaerobic inner part of the piles more methane, increasing climate-altering gas emissions.
Moreover, because SMX and CIP decreased substantially in digestate, the latter is more desirable as
an organic fertilizer in order to reduce for agricultural soils antibiotic contamination and
presumably antibiotic resistant genes.
Finally, the antibiotics influenced the structure of the microbial community since Bacteria increased
and Archaea decreased with the rise in antibiotic concentrations. In particular, at the end of the
experiment, a predominance of Archaea (Methanobrevibacter and Methanosphera), effective in the
production of methane along the hydrogenotrophic metabolic pathway, was observed. In contrast,
the presence of antibiotics counteracted the functionality of acetotrophic methanogens, as confirmed
by the acetic acid in-creased concentrations detected until the end of the experiment”.
Consider emphasizing: It seems that SMX impairs the metabolic activity of Archaea more than CIP.
An unexpected higher methane value for mix10 than mix5 (Figure 1) results from higher SMX
degradation (Figure 3), yet it does not correspond well to the total and relative abundance of
Archaea. However, the key factor for the utilization of H2 and production of methane may be the
dominance of Methanosphaera archaeon when the antibiotics were partially degraded.
Thank you for your keen observation: we have added the following sentences “Some aspects remain
to be clarified concerning the role of SMX in the mixtures. Indeed, at the end of the experiment,
Mix_10 produced more CH4 than Mix_5 (Figure 1), in line with the removal rates (Figure 3). On
the other hand, SMX_5 produced CH4 faster than CIP_5, since the production plateau is reached at
days 22 (SMX_5) and 40 (CIP_5), showing a lower effect on the microbial community of SMX
than CIP”
(lines 198-203).
Specific remark
Line 86 change to: are shown in Figure 1 or were observed (Figure 1)
Thank you for your useful observation. We have changed in the text with “were observed (Figure
1).
Line 92-93 consider changing to: The start of CH4 production, corresponded to the H2 decrease in
the biogas on the timelines (Figure 1)
Thank you for your suggestion. We have changed the text with your suggestion.
Line 95 Did you mean the lowest and highest value in the period of 10-20 days? The figure indicate
they are not Control and CIP_5, respectively.
Thank you for your observation. We have corrected this oversight. In fact, the higher value refers to
MIX_5 and we have added it in the text
Line 100 add also a value for the control.
Thank you. We have added a value for the control.
Line 109 add data of SMX alone in Figure 1 as you did for Figure 3.
Thank you. We have added data of SMX alone in Figure 1.
Line 113 which in my opinion, results from a higher degradation rate of SMX
Thank you for your fair observation. This result confirms a lower impact of SMX on the AD
microbial community than CIP, also due to its higher degradability.
Line 184 mix 5
Thank you. We have corrected this in the text.
Line 223 colours of bars are mixed for different variants. Why does CIP5 lack 3 and 27-day
measurements and mix 2,5 lack 3-day measurements? Check table 1.
Thank you very much. You are right. We have updated Figure 4a in accordance with your
observations.
Line 280 …(Figure 6) showed …
Thank you, done.

Reviewer 3 Report
Manuscript ID: antibiotics-1825556 by Mazzurco Miritana et al., entitled “Effects of ciprofloxacin alone or in mixture with sulfamethoxazole on the efficiency of anaerobic digestion and its microbial community" investigates the impact of antibiotics on microbial community functionality in anaerobic digestion (AD) process of cattle manure.
General remarks:
The proposed manuscript is well written and provides new insight into the possible factors affecting biogas production. Antibiotics are often used in excess in modern livestock farms but their subsequent impact on other environmental processes is often overlooked. Here I must disagree with the authors that “AD proved to be a valuable treatment for reducing residual concentrations of CIP and SMX antibiotics from livestock manure”. Your results concerning the removal of antibiotics (Figure 3) (on average 6% and 30% for CIP and SMX, respectively) as well as literature data indicate that antibiotics have a strong impact on microbiomes. In my opinion, you must remove this statement from conclusions and add at least a short discussion that antibiotics have a higher impact on acetoclastic methanogens than hydrogenotrophic (reference 1 from below), and considering hydrogenotrophic methanogens occurrence in environmental samples (2) the remaining antibiotics may have a relatively low impact on natural biogas release but it needs to be highlighted that antibiotics can impair acetate utilization and many other biological processes, promote the transfer of antibiotic resistance genes to indigenous environmental bacteria and Archaea (3).
11. (9 from your reference list) Cheng, D.L.; Ngo, H.H.; Guo, W.S.; Chang, S.W.; Nguyen, D.D.; Kumar, S.M.; Du, B.; Wei, Q.; Wei, D. Problematic effects of 487 antibiotics on anaerobic treatment of swine wastewater. Bioresource Technology 2018, 263, 642–653, 488 doi:10.1016/J.BIORTECH.2018.05.010
22. Pyzik, A., Ciezkowska, M., Krawczyk, P. S., Sobczak, A., Drewniak, L., Dziembowski, A., & Lipinski, L. (2018). Comparative analysis of deep sequenced methanogenic communities: identification of microorganisms responsible for methane production. Microbial cell factories, 17(1), 1-16. https://doi.org/10.1186/s12934-018-1043-3
33 . Xu, R., Yang, Z. H., Zheng, Y., Wang, Q. P., Bai, Y., Liu, J. B., ... & Fan, C. Z. (2019). Metagenomic analysis reveals the effects of long-term antibiotic pressure on sludge anaerobic digestion and antimicrobial resistance risk. Bioresource technology, 282, 179-188. https://doi.org/10.1016/j.biortech.2019.02.120
Consider emphasizing: It seems that SMX impairs the metabolic activity of Archaea more than CIP. An unexpected higher methane value for mix10 than mix5 (Figure 1) results from higher SMX degradation (Figure 3), yet it does not correspond well to the total and relative abundance of Archaea. However, the key factor for the utilization of H2 and production of methane may be the dominance of Methanosphaera archaeon when the antibiotics were partially degraded.
Specific remark
Line 86 change to: are shown in Figure 1 or were observed (Figure 1)
Line 92-93 consider changing to: The start of CH4 production, corresponded to the H2 decrease in the biogas on the timelines (Figure 1)
Line 95 Did you mean the lowest and highest value in the period of 10-20 days? The figure indicate they are not Control and CIP_5, respectively.
Line 100 add also a value for the control.
Line 109 add data of SMX alone in Figure 1 as you did for Figure 3.
Line113 which in my opinion, results from a higher degradation rate of SMX
Line 184 mix 5
Line 223 colours of bars are mixed for different variants. Why does CIP5 lack 3 and 27-day measurements and mix 2,5 lack 3-day measurements? Check table 1.
Line 280 …(Figure 6) showed …
Author Response
Were the samples used for the AD setup antibiotic free? This should be mentioned as any present
could effect the test samples (potential synergistic effects).
Thank you for your comment. Yes, the ingestate was initially analysed for SMX and CIP residual
concentrations. SMX was 0.30 mg/L and CIP 0.1 mg/L.
We have added this information in the manuscript: “The bottles were transported to the laboratory
where ingestate was analysed to detect possible residual amounts of SMX (0.3 mg L -1 ) and CIP (0.1
mg L -1 ) before setting up the experiment (lines 325-326).
Figure 1 is very crowed and this makes it very difficult to see the values and understand what is
going on. I would suggest dividing it into figure 1a and 1b splitting into CH4 and H2
Thank you for your suggestion. Unfortunately, CH4 and H2 need to be in the same figure in order
to highlight that methane production starts in correspondence of hydrogen decrease. Really, we
have tried to divide the figure, but it was not clear in this way. In any case, we have improved the
figure. In any case we have improved the figure and the legend label in order to make more clear
H2 (in the left side) and then CH4 production (in the right side).
In Figure 2 the Figure key is way too big and the actual figures way too small.
Thank you. We have improved it
In line 312 what does the number 15 denote? Days? Weeks,
Thank you very much. 15’ means 15 minutes (min), we have specified this in the text.

Round 2
Reviewer 1 Report
After all the edits, looks good to me!